# A Study on the Impact of Organizing Environmental Awareness and Education on the Performance of Environmental Governance in China

**DOI:** 10.3390/ijerph191912852

**Published:** 2022-10-07

**Authors:** Yifei Niu, Xi Wang, Ciyun Lin

**Affiliations:** 1College of the Humanities, Jilin University, Changchun 130012, China; 2Economic Research Institute, Jilin Academy of Social Sciences, Changchun 130033, China; 3School of Transportation, Jilin University, Changchun 130022, China

**Keywords:** environmental governance, environmental education and publicity, public participation, environmental governance performance

## Abstract

The advancement of technology and economic development has raised the standard of living and at the same time brought a greater burden to the environment. Environmental governance has become a common concern around the world, and although China’s environmental governance has achieved some success, it is still a long way from the ultimate goal. This paper empirically analyzes the impact of environmental publicity and education on environmental governance performance, using public participation as a mediator. The results show that: the direct effect of environmental publicity and education on environmental governance performance is not significant; environmental publicity and education have a significant positive effect on public participation; public participation significantly contributes to environmental governance performance; public participation shows a good mediating effect between environmental publicity and education and environmental governance performance. The government should adopt diversified environmental protection publicity and education in future environmental governance, and vigorously promote public participation in environmental governance so that the goal of environmental governance can be fundamentally accomplished by all people.

## 1. Introduction

With the progress of economic and social development, China’s urbanization is accelerating, and the increase in urban population has aggravated the deterioration of environmental pollution problems. With the rapid economic development and shortening of product life cycles, the rate of product replacement has become faster and faster, and the global energy shortage has caused increasing concern for environmental protection in various countries [1,2,3]. China’s current ecological and environmental situation is very serious due to overpopulation, the long-term unreasonable exploitation of resources such as land, forests, water, and minerals, and the lack of necessary protection and construction of the ecological environment. At present, the deterioration of China’s ecological environment is summarized roughly as follows.

(1) Serious water and soil erosion. Soil erosion in China continues to show a “double decline” in area intensity and a “double reduction” in water and wind erosion. In 2021, the national soil erosion area will be 2,674,200 square kilometers, down 274,900 square kilometers from 2011, the percentage of strong and above grade will drop to 18.93%, and the soil and water conservation rate will reach 72.04% [4,5]. (2) The area of desertified land is expanding. The national desertification land area has reached 2.62 million square kilometers, accounting for 27.3% of the national land area, expanding at a rate of about 2460 square kilometers per year. (3) The area of degraded, sandy, and alkaline grassland is increasing year by year. The area of trivialized grasslands has reached 135 million hectares and is increasing at the rate of 2 million hectares per year [6]. (4) The area of acid rain area is further expanded, and the degree is worsening. Serious air pollution, mainly of the soot type, leads to a large area of acidic precipitation, and the area of acid rain area in China is about 466,000 square kilometers accounting for 4.8% of the national land area [7,8]. (5) The quality of the water environment is deteriorating, and water pollution accidents are frequent. According to incomplete statistics, in 2021, the country’s 458 daily discharge of direct sea pollution sources was more than 100 tons, the total volume of sewage discharge was about 7.28 billion tons, and they reported the largest integrated outfall emissions, followed by industrial sources of pollution, consequently, water shortage will be more serious [9,10]. (6) With the development of society, noise pollution has become an important source of pollution that affects people’s physical and mental health [11,12,13,14,15,16,17]. Noise pollution is recognized by the scientific community as an environmental pollutant related to sleep disorders and learning disabilities. Studies have shown that long-term exposure to highways, railroads, airports, and recreational noise can reduce work performance, make people irritable, and can seriously cause hypertension, heart attacks, and other diseases, which is also one of the sources associated with air pollution [18,19,20,21,22,23,24,25].

National governments are paying more and more attention to environmental protection and ecological civilization construction work. To this end, they keep formulating and improving relevant laws and regulations, however, environmental protection is not only the task of the state and government, but also the responsibility and obligation of every resident, and long-term environmental protection and ecological civilization cannot be built without public participation. Under the constraints of environmental governance [26], environmental education can effectively improve people’s awareness of environmental protection [27,28,29]. Global environmental governance is the fundamental way to solve the human environmental crisis, and environmental awareness and education have been considered tools to help alleviate environmental problems [30,31]. Under the strong leadership of the Party Central Committee with Comrade Xi Jinping at the core, China’s higher education has moved with the times, built the world’s largest higher education system, cultivated a large number of high-quality specialists, and played an extremely important role in national revitalization, economic construction, social development, and scientific and technological progress. Historic achievements and changes in the pattern of higher education have been made. The world’s largest higher education system has been built, with the total number of students enrolled exceeding 44.3 million, and the gross enrollment rate of higher education has increased from 30% in 2012 to 57.8% in 2021, an increase of 27.8 percentage points, achieving a historic leap. Thus, higher education has entered a stage of universalization recognized worldwide. With 240 million people receiving higher education and an average of 13.8 years of education for the new workforce, the quality and structure of the workforce have undergone significant changes and the quality of the entire nation has been steadily improved. The awakening of public environmental awareness means that citizens gradually have a correct understanding of the relationship between human beings and nature in line with the essence of ecological civilization, and this awakening of environmental awareness needs to be inspired by education, and the process of gradually awakening environmental awareness is the process of gradually cultivating citizens into ecological citizens. Environmental education is the key to cultivating eco-citizens, and it has become an important value of environmental education in the context of ecological civilization construction to enhance citizens’ awareness of ecological civilization through environmental education. The government’s existing legislative practice of environmental education reflects the importance it attaches to the cultivation of ecological citizens through its environmental education training and other systems, i.e., the introduction of special legislation on environmental education. The awakening of public environmental consciousness requires the introduction of special legislation on environmental education to clarify the systems of ecological citizenship cultivation at the central level and effectively guarantee the effective development of ecological citizenship cultivation at the society-wide level, so as to promote the continuous awakening of citizens’ environmental consciousness and make all citizens in society become the main body in line with environmental protection and ecological civilization construction.

Liu P. pointed out in his study that perfect environmental protection propaganda and education work helps to enhance public participation in the construction of ecological civilization and protects the environment while safeguarding public interests; only by achieving universal environmental protection can we truly achieve long-term ecological civilization construction and the great rejuvenation of the Chinese nation [32]. Fan Y.W. proposed corresponding strategies and measures by analyzing the complexity and forms of urban environmental governance in China, in which he proposed that the background of China’s rapid social and economic development hides a severe form of environmental governance, and the government should actively seek the support of enterprises, environmental organizations, and popular social forces while committing to sound policies and regulations and strict compliance, through more non-governmental forces’. In addition, public participation is an effective way to improve the efficiency of environmental protection, and through various forms of environmental protection publicity and education, the public’s environmental awareness and quality can be improved comprehensively, and the citizens’ legitimate right to supervision and information can be guaranteed [33]. Ecological environmental governance requires the collaborative participation of the government and the public, and that the two work together to provide services for environmental governance to achieve the optimization of the governance process and results. Public participation in environmental governance generates competent evaluations of external information perceptions based on their consciousness, among which the perceived evaluations of environmental risks, government performance, and self-efficacy all feature. The disclosure of bad environmental information can enhance the public’s perception of environmental risks, the threat of environmental pollution to their lives and the damage to their interests, consequently, stimulating the public’s defense mentality. Thus, introducing environmental education (especially, environmental health) could be used as one solution [34], promoting public participation in environmental governance, and good government environmental treatment performance can enhance the public’s satisfaction and recognition of government work, and increased satisfaction can enhance the public’s participation in environmental governance. The public’s participation in environmental governance has an obvious subjective motivation, and a good perception of self-efficacy can enhance the public’s enthusiasm to participate in environmental governance and help improve the performance of environmental governance [35,36,37].

In summary, environmental protection publicity and education can improve public perceptions of environmental protection, enhance public awareness of environmental protection, and promote public participation in environmental governance, and the government and the public can work together to serve environmental governance to maximize environmental governance performance [38,39,40]. Given the above literature findings, this paper takes public participation in environmental governance as an entry point to analyze in depth the impact of organizing environmental protection publicity and education on environmental governance performance and makes suggestions from the perspective of promoting public participation in environmental governance.

## 2. Research Hypotheses

### 2.1. Impact of Environmental Awareness and Education on Environmental Governance Performance

The Third Plenary Session of the 18th Central Committee of the Communist Party of China (CPC) introduced policies and measures to modernize the governance system and governance capacity. As an important part of the modernized governance system, environmental governance, together with politics, economy, culture, society, and ecology, forms the national governance system. To fundamentally solve the task of urban environmental governance, it is necessary to create a situation of multi-governance, through environmental information disclosure and environmental education, to protect the legitimate rights and interests of citizens while enhancing the environmental awareness of social organizations and citizens. It is also helpful to improve the effectiveness of government environmental governance. It has been pointed out that the transparency and openness of environmental information have a significant impact on the performance of environmental governance and given the complexity, professionalism, and a certain degree of closure of environmental information, the right to information of citizens, enterprises, and social organizations is not fully guaranteed, which has a negative impact on the effectiveness of governmental environmental monitoring and governance. With the advancement of technology and the development of innovative environmental governance models, management mechanisms with high transparency and openness have become a new environmental governance model, which combines environmental information disclosure with environmental education to restrain corporate behavior, promote the participation and supervision of citizens and social organizations in environmental governance, and collaboratively improve government environmental governance performance [41,42,43]. Environmental information disclosure and publicity and education are complementary measures to environmental governance, and their actual effects are directly related to the strength of disclosure and publicity. Environmental information disclosure enhances the pressure faced by emission enterprises from public opinion and regulatory authorities, prompting them to increase the disclosure of environmental information, improve and upgrade their environmental protection technologies, and take up the environmental protection tasks they should accomplish. Environmental information disclosure combined with environmental publicity and education enhances the success of these efforts. The combination of environmental information disclosure and environmental education enhances public scrutiny of the effectiveness of environmental management by government departments and creates a multi-party accountability situation for local governments from higher levels of government, the public, and media opinion, prompting local governments to effectively implement environmental management [44,45,46]. In general, through environmental protection publicity and education, the supervision and pressure on emission enterprises and government departments through various aspects such as social organizations, citizens, and media public opinion have promoted the strength of environmental governance by government departments. Based on the above analysis, this paper proposes the following hypotheses.

**Hypothesis** **1** **(H1).***Environmental protection publicity and education positively affect environmental governance performance*.

### 2.2. The Impact of Environmental Publicity and Education on Public Participation

With the continuous improvement of living standards and education, although the environmental awareness of our residents has been greatly enhanced, there are still some shortcomings. The pilot implementation of many years of waste separation action at present the overall effect is poor, and the behavior of randomly throwing away garbage is still very common, the government vigorously promotes environmentally friendly consumer behavior also has little effect, “white pollution The problem of “white pollution” still exists. Industrial pollution caused by China’s rapid urbanization and industrialization is a serious threat to the ecological environment and human health, which stimulates public appeals for better environmental quality and public participation in regional environmental governance [47]. This shows that public awareness of environmental protection needs to be improved. Environmental protection publicity and education activities should reach out to the public and target outstanding environmental protection problems with the right remedy for publicity and education. On the one hand, popular publicity and education on environmental protection knowledge can be conducted in communities, schools, enterprises, cities, and villages to deepen public awareness of environmental protection and call for universal participation in environmental protection; on the other hand, thematic publicity and education activities can be held in regional water quality warning stations, air monitoring stations and motor vehicle, on the other hand, education activities can be held in regional water quality warning stations, air monitoring stations and motor vehicle exhaust testing centers, etc., with relevant professionals explaining environmental protection knowledge and issues that are closely related to the public [48]. Rural household waste management is an important part of building beautiful villages and achieving the goal of ecological livability, and the willingness of rural residents to participate in household waste management is influenced by a variety of factors such as economic conditions, environmental awareness, and regulations. The current level of participation in waste management initiatives is low overall. Some scholars used village cadres as an entry point to study and analyze the impact of environmental education on rural residents’ willingness to participate in domestic waste management, and the results showed that environmental education based on the moderating role of village cadres had a significant impact on rural residents’ willingness to participate in domestic waste management. Therefore, the government should carry out diversified methods of publicity and education to enhance residents’ environmental awareness and willingness to participate in environmental management, taking into account the actual needs [49]. Some studies have pointed out that environmental protection publicity and education is the guidance and interaction between society and public participation in environmental governance, and environmental protection is working closely related to everyone’s interests, and any social organizations and individuals have the obligation and responsibility to protect the environment, and through environmental protection publicity and education, the public is encouraged to participate in the supervision and management of environmental governance, and everyone, regardless of occupation or position, reflects their attitude and choice of environmental protection, and the whole population will be governed to completely solve environmental problems [50,51,52]. In summary, through various forms of environmental protection publicity and education, we can deepen the public’s understanding and awareness of environmental protection, make the public realize the connection between environmental protection and their own interests, promote public participation in environmental governance from both subjective awareness and objective needs, and enhance the willingness to participate [53]. Based on the above analysis, this paper proposes the following hypotheses.

**Hypothesis** **2** **(H2).***Environmental protection publicity and education enhance the public’s willingness to participate*.

### 2.3. Mediation Effect of Public Participation

Public participation in environmental governance not only refers to subjective environmental protection behaviors but more importantly, it is a variety of ways to monitor and put pressure on the government’s environmental governance work such as letters and visits, social organization intervention, litigation, and creating public opinion; environmental protection letters and visits, environmental protection litigation and social organization intervention can provide clues for the government’s environmental governance while creating public opinion creates greater pressure on the government’s environmental protection behaviors and enhances the public opinion has become a key factor in environmental governance in recent years [54,55,56,57]. In recent years, the public has become an important force in environmental governance, and the ways and means of public participation in environmental governance have been diversified, such as environmental petitions, environmental public opinion, CPPCC proposals, NPC motions, etc. Among them, environmental petitions reflect a more significant promotion effect on environmental governance performance than environmental public opinion, while the promotion effect of CPPCC proposals and NPC motions is not obvious, and the reason for this is that one of the main subjects of environmental governance, to avoid economic and social losses caused by environmental mass events caused by mishandling, so public demands are more important to local governments, and public environmental petitions can bring more pressure than CPPCC proposals and NPC motions, so the impact on environmental governance performance is greater [58,59,60]. Based on the interprovincial panel data from 2011–2015, some scholars constructed a model to empirically analyze the effects of three public participation channels, namely, complaint petitions, proposals, and self-media opinion, on regional environmental governance performance, and the results showed that public participation has a positive contribution to regional environmental governance performance, with the positive contribution effect of self-media opinion being the most significant. Based on the above literature, it can be seen that the ways of public participation in environmental governance are diversified and the effects of different ways on environmental governance performance vary, but promoting public participation in environmental governance through environmental education can indirectly contribute to the improvement of government environmental governance performance. Based on the above analysis, this paper proposes the following hypotheses.

**Hypothesis** **3** **(H3).***Public participation can play a mediating effect between environmental publicity and education and environmental governance performance*.

## 3. Research Design

### 3.1. Data Sources

This paper used questionnaires to complete the data collection and acquisition. We randomly distributed questionnaires in townships, urban communities, enterprises, government environmental protection departments, and other related departments, the research subjects were township and urban community residents, enterprise middle-level and above managers, government environmental protection department staff and responsible person, government-related department staff and responsible person, etc., over 6 months, a total of 1200 questionnaires were distributed. After eliminating invalid questionnaires, a total of 1017 valid questionnaires were collected, with a valid recovery rate of 84.8%.

The content of the questionnaire was mainly for environmental protection publicity and education, public participation, and government environmental governance information survey. Each aspect contained a number of questions, using a five-level rating system to evaluate the valuation of each question, corresponding to completely not meet, not meet, not sure, meet, fully meet, respectively, assigned to the value of 1–5, environmental protection publicity and education, public participation and government environmental governance rating for the corresponding questions. The scores for environmental education, public participation, and government environmental governance were the sum of the corresponding questions [61,62,63,64]. The detailed questionnaire scale is shown in Table 1.

In order to ensure the credibility and validity of the questionnaire data, the reliability and validity of the scale need to be checked before the formal analysis, using Cronbach’s α reliability coefficient method and CITI value to test the data reliability, with α > 0.7 and CITI > 0.5 as the internal consistency of the questionnaire qualified, that is, the questionnaire has good stability and reliability, reliability test details The results are shown in Table 2, the CITI values are greater than 0.5, and the reliability of all four variables is greater than 0.7, indicating that the questionnaire passed the reliability test.

SPSS23.0 statistical software (IBM, New York, NY, USA) was used to conduct KMO and Bartlett sphere tests to verify the validity of the scale items, and the results showed that the KMO value was 0.966 and the Bartlett sphere test’s approximate chi-square value was 6139.573 (*p* < 0.001), indicating that the scale items had good structural validity of cut elements in line with the conditions of factor analysis. The results of convergent validity and discriminant validity analysis of the sample data are shown in Table 3 and Table 4. The data in the tables show that the factor loadings of the items are greater than 0.5, the CR values of the four variables are greater than 0.7, and the AVE values are greater than 0.5. The convergent validity and discriminant validity of the questionnaire data were good.

### 3.2. Model Design

According to the analysis of the aforementioned literature, environmental protection publicity and education has a certain degree of positive impact on environmental governance performance, but it mainly has an indirect effect on environmental governance by influencing the will and ways of public participation in environmental governance, which has a direct impact on environmental governance. Based on the analysis of literature results, the theoretical model designed in this paper is shown in Figure 1.

### 3.3. Model Test

The Mplus software was used to test the goodness-of-fit indicators of the structural model, and the test results are shown in Table 5, from which it can be seen that all seven goodness-of-fit indicators meet the critical value requirements, indicating that the constructed model has good goodness-of-fit.

## 4. Empirical Analysis

### 4.1. Path Analysis

Based on the constructed theoretical structure model, and based on the sample data, Mplus software was applied to calculate the paths and impact relationships between environmental education, public participation, and environmental governance performance. The detailed results shown in Table 6 are not fully valid, the reason for this analysis is that the government, as one of the main participants in environmental publicity and education, organizes environmental publicity and education for the fundamental purpose of encouraging enterprises and the public to participate in environmental governance, and the direct effect of this on environmental governance performance is not significant. Both environmental publicity and education show a significant positive effect on public participation and public participation on environmental governance performance, indicating that environmental publicity and education can effectively promote public participation in environmental governance, and public participation in environmental governance significantly improves environmental governance performance.

### 4.2. Mediating Effect Analysis

The results are shown in Table 7, which shows that the confidence intervals of both the specific mediating effect and the total mediating effect are greater than 0, indicating that the mediating effects of public participation intention, public participation channel and This indicates that the mediating effects of public participation, public participation channels and public participation in environmental education and environmental governance performance are all valid, which verifies the hypothesis of this paper.

## 5. Conclusions and Discussions

This paper investigated the impact of environmental education on environmental governance performance, using public participation as the entry point for studying the mediating effects of environmental governance. The empirical results show that environmental publicity and education have a positive effect on environmental governance performance to some extent but are not significant. Environmental publicity and education have a significant positive effect on both public participation willingness and allocated resources; both public participation willingness and public participation significantly contribute to the improvement of environmental governance performance. Public participation shows a good mediating effect between environmental publicity and education and environmental governance performance [42,43].

Aldo Leopold, a famous American ecologist and founder of the land ethic, once said the essence of resource protection did not lie in a few government engineering programs, but fundamentally, in a change in the consciousness of all people. To protect the ecological environment, we should not only rely on the macro-control of government departments, and increase investment in environmental protection to take the path of sustainable development, but more importantly, we should improve the quality of awareness in the whole population, increase environmental publicity, and enhance environmental awareness. Environmental propaganda should adhere to the purpose of improving the environmental awareness of all sectors of society and the general public, decision-makers, factories and mining enterprises, young people, and society as a whole. All departments should make full use of the media and widely popularized scientific knowledge of environmental protection. Using close-to-life, reality-based publicity activities can enhance the nation’s environmental awareness and understanding of the legal system, leading to improvements in the public consciousness and increased participation in environmental action. Only in this way can we lay a good foundation for the smooth implementation of various policies and measures. Higher environmental awareness is a sign of progress in social civilization, and the higher the environmental awareness, the lower the resistance encountered to the implementation of environmental policies. Usually, it is necessary to continuously strengthen environmental publicity and education, and it is also possible to raise public environmental awareness by expanding the environmental rights and interests enjoyed by society, including the right to environmental supervision, the right to environmental information, the right to environmental claims, and the right to environmental discussion. In addition, the strengthening of environmental awareness must be accompanied by the construction of a supporting legal system, and the establishment of a virtuous ecological cycle by giving equal importance to ecological exploitation and protection and restoration as guidelines. Based on the theoretical results of the literature and combined with the results of the empirical analysis, this paper puts forward the following recommendations.

First, extensive participation in publicity and education. It is difficult for government-led publicity and education to cover all areas, so it is necessary to use the influence of corporate organizations and celebrities to expand the scope of environmental publicity and education. The cause of environmental protection concerns the vital interests of everyone; any social organization or individual has an obligation to take responsibility for protecting the environment and preventing pollution, so the ambitious goal of ecological livability and sustainable development cannot be achieved without joint efforts of all people and society as a whole.

Second, encourage public participation in environmental governance. The government should also take the lead in encouraging the public to participate in environmental management, and at the same time mobilize non-government forces to encourage public participation in environmental management. The government’s environmental management efforts will create a favorable social atmosphere. In addition, we should give full play to the role of private environmental organizations, guide and encourage them to participate in environmental governance, promote public supervision and media attention, and create a strong social force for environmental governance to comprehensively promote the goal of green development, which is also of great significance to China’s strategic goal of sustainable development.

Third, improve the relevant laws and regulations. The implementation of environmental protection policies cannot be implemented without the guarantee of laws and regulations, and perfect laws and regulations also have a direct impact on the operability of the relevant policies. We need to strengthen the establishment and implementation of laws and regulations related to environmental protection and rely on a sound legal system to comprehensively promote environmental governance. Scientific management systems, standardized operation processes, and strong supervision mechanisms are the important objective conditions for public participation in environmental governance. The environmental emergency management mechanism, media opinion monitoring mechanism, and environmental information disclosure mechanism need to be further supplemented and improved. A sound legal system can also provide the public with clear environmental rights, raise public awareness of environmental protection, and make the public participate more actively in environmental protection work. In short, a sound legal system is an important guarantee for public participation in environmental governance, and an important way to improve environmental governance performance.

Forth, improve the efficiency of government feedback. The government plays a role in guiding and interacting with the public in the process of environmental governance. Mass environmental incidents are typically generative of common demands for environmental governance from the public. If such demands do not receive timely and effective feedback from the government, it is likely to lead to a decrease in the public’s trust in the government and even trigger a crisis of government credibility. A positive interaction cycle between the public and the government is an important guarantee for the smooth development of the government’s environmental protection business, so the government should actively give feedback when the public clearly expresses their demands and give clear responses and answers to ensure smooth and harmonious communication between the two. In addition, the government environmental protection department should regularly sort out the statistics of public demands, actively solicit suggestions and opinions from the public, and make timely improvements in response to the shortcomings of the work.

## Figures and Tables

**Figure 1 ijerph-19-12852-f001:**
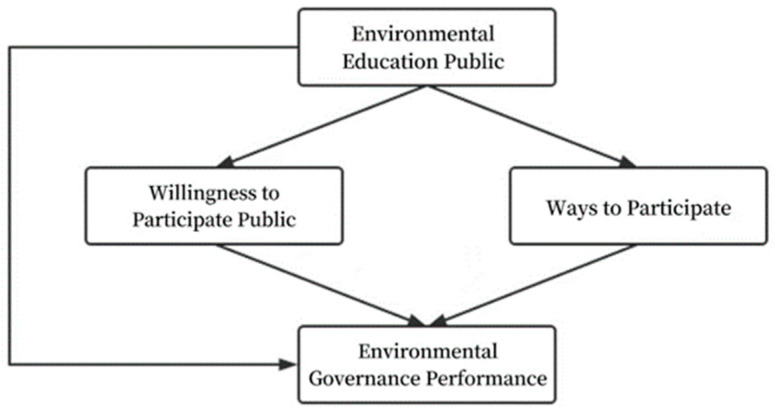
Theoretical structural model.

**Table 1 ijerph-19-12852-t001:** Survey questionnaire scale design.

Variables	Option Description	Number
Environmental awareness and education	Distribute brochures	A_1_
	Conducting knowledge lectures	A_2_
	Environmental Information Disclosure	A_3_
	Conducting special events	A_4_
Willingness to participate	Active attention to environmental information	B_1_
	Active feedback on adverse environmental information	B_2_
	Actively respond to government environmental protection policies	B_3_
Ways to participate	Environmental protection letters and petitions	C_1_
	Create media opinion	C_2_
	Telephone network complaints	C_3_
	Suggestions to the environmental protection department	C_4_
Environmental Governance Performance	Increased number of environmental enforcement cases	D_1_
	The number of NPC and CPPCC proposals increased	D_2_
	Increase in total investment in environmental governance	D_3_

**Table 2 ijerph-19-12852-t002:** Results of the reliability test of the questionnaire.

Variables	Title Item	CITI	Cronbach’s α
Environmental awareness and education	A_1_	0.728	0.965
	A_2_	0.681	
	A_3_	0.767	
	A_4_	0.776	
Willingness to participate	B_1_	0.614	0.796
	B_2_	0.801	
	B_3_	0.771	
Ways to participate	C_1_	0.774	0.876
	C_2_	0.703	
	C_3_	0.873	
	C_4_	0.836	
Environmental Governance Performance	D_1_	0.771	0.883
	D_2_	0.728	
	D_3_	0.681	

**Table 3 ijerph-19-12852-t003:** Results of convergent validity test.

Variables	Title Term	Factor Loadings	Unstandardized Coefficient	Standard Error	*t*-Value	CR	AVE
Environmental awareness and education	A_1_	0.769	1.041	0.073	12.461	0.932	0.529
	A_2_	0.767	1.167	0.08	13.532		
	A_3_	0.751	0.945	0.073	13.478		
	A_4_	0.744	1.047	0.094	12.495		
Willingness to participate	B_1_	0.736	1.158	0.089	12.873	0.862	0.611
	B_2_	0.733	1.152	0.06	13.186		
	B_3_	0.805	0.956	0.069	14.798		
Ways to participate	C_1_	0.767	1.236	0.078	16.565	0.791	0.662
	C_2_	0.755	1.338	0.059	13.407		
	C_3_	0.829	1.035	0.068	14.870		
	C_4_	0.828	1.041	0.066	16.465		
Environmental Governance Performance	D_1_	0.805	1.225	0.100	15.520	0.803	0.543
	D_2_	0.726	1.136	0.070	13.422		
	D_3_	0.716	1.230	0.071	15.059		

**Table 4 ijerph-19-12852-t004:** Results of the discriminant validity test.

Variables	A	B	C	D
A	0.736			
B	0.580 **	0.729		
C	0.473 **	0.423 **	0.716	
D	0.447 **	0.366 **	0.509 **	0.828

Note: ** indicates *p* < 0.01; diagonal values in the table are the square root of the AVE of the corresponding variable.

**Table 5 ijerph-19-12852-t005:** Structural model goodness-of-fit index measures.

Indicators	Statistical Value	Test Critical Value	Superiority Test
X^2^/df	2.337	<3	Pass
CFI	0.95	>0.9	Pass
GFI	0.934	>0.9	Pass
NFI	0.993	>0.9	Pass
IFI	0.975	>0.9	Pass
TLI	0.965	>0.9	Pass
RMSEA	0.135	<0.5	Pass

**Table 6 ijerph-19-12852-t006:** Path analysis.

Paths	Standardization Factor	Non-Standardized Factor	SE	*p*
Environmental awareness and education → Environmental governance performance	0.071	0.111	0.041	0.159
Environmental awareness and education → public willingness to participate	0.644	0.719	0.075	0.001
Environmental education → Public participation channels	0.522	0.623	0.098	0.001
Public willingness to participate → Environmental governance performance	0.477	0.425	0.052	0.001
Public Participation Pathways → Environmental Governance Performance	0.425	0.397	0.029	0.001

**Table 7 ijerph-19-12852-t007:** Intermediary effect test.

Paths	Effect	Standardization Factor	Non-Standardized Coefficient	*p*	95% Confidence Interval	
					Lower Limit	Upper Limit
Environmental awareness and education → Environmental governance performance	Direct effect	0.071	0.111	0.148	0.060	0.211
Environmental awareness and education → public willingness to participate → environmental governance performance	Specific mediating effects	0.303	0.210	0.000	0.162	0.313
Environmental awareness and education → Public participation pathways → Environmental governance performance		0.142	0.116	0.000	0.162	0.219
Environmental awareness and education → public participation → environmental governance performance	Total intermediation effect	0.562	0.436	0.000	0.422	0.629
Environmental awareness and education → Environmental governance performance	Total effect	0.632	0.547	0.000	0.507	0.773

## Data Availability

Not applicable.

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
