# Peer review of "A Study on the Impact of Organizing Environmental Awareness and Education on the Performance of Environmental Governance in China"

_ijerph, 2022, doi:10.3390/ijerph191912852_

Round 1
Reviewer 1 Report
Authors analysed very specific country as China, where the strongest power has the government. Therefore, I do not understand the provided model because government influence environmental education. Can authors more discuss the logic of provided model and the specification of analysed country. It is also not clear how the scales were constructed, because it absolutely not enough to ask two questions which reveal the environmental governance performances. So, the scales should be justified. It is also not clear for me whether the performed survey was representative because there is no information about the respondents. The descriptive analysis of analysed factors also should be provided, for example what level of environmental awareness and education is in China. However, primarily authors should justify the provided model and analysed variables. The discussion section also should be more expanded. For example, what are the factors which influence environmental education and how government can contribute to it.
Author Response
Response to Reviewer 1 Comments
Point 1: Authors analysed very specific country as China, where the strongest power has the government. Therefore, I do not understand the provided model because government influence environmental education. Can authors more discuss the logic of provided model and the specification of analysed country.
Response 1: In conducting the analysis, this paper investigates the impact of organizing environmental awareness and education on environmental governance performance, using public participation as a medium to empirically analyze the impact of environmental awareness and education on environmental governance performance, using China as an example. The government does play a significant role in raising the environmental awareness of citizens. The logic and basis of the analysis established in the structural model of this paper is that environmental advocacy education has a positive impact on environmental governance performance to some extent, but mainly indirectly on environmental governance by having an influence on the willingness and means of public participation in environmental governance, while public participation has a direct impact on environmental governance, so we established a theoretical structural model based on this, and conducted model testing and empirical analysis.
Point 2: It is also not clear how the scales were constructed, because it absolutely not enough to ask two questions which reveal the environmental governance performances. So, the scales should be justified.
Response 2: In response to your question about how the scale is constructed, our questionnaire is designed to investigate information on three aspects: environmental education, public participation, and government environmental governance, each of which contains multiple questions. The scores for publicity and education, public participation, and government environmental governance are the sum of the scores for the corresponding question items. Our team created the scale based on repeated references from previous readings of the references, and the corresponding reference citations we have marked in the article and highlighted in red font.
Point 3: It is also not clear for me whether the performed survey was representative because there is no information about the respondents.
Response 3: In response to your question about the lack of respondent information, we used questionnaires for the study of township and urban community residents, middle-level and above managers of enterprises, government environmental protection department staff and responsible persons, government-related department staff and responsible persons, etc., over a period of 6 months, a total of 1200 questionnaires were distributed, after eliminating invalid questionnaires, a total of 1017 valid questionnaires were collected, with a valid The effective recovery rate is 84.8%. The comprehensive comparative analysis of the recovered effective questionnaires shows that the data is sufficient and the effective recovery rate of the questionnaires is high, which we think is representative.
Point 4: The descriptive analysis of analysed factors also should be provided, for example what level of environmental awareness and education is in China. However, primarily authors should justify the provided model and analysed variables.
Response 4: In response to your question about adding a descriptive analysis of how the environment and education levels are in China, our team has taken your suggestion fully into account and revised it in full detail. We have added the aspect of environmental pollutants to the article and specifically described the current status of China in terms of each category of environmental pollutants, cited the corresponding references, and described the environmental awareness of the people and proposed the corresponding countermeasures. In terms of education level, the current form of education in China is fully described, and the corresponding data and analysis are provided, and the corresponding references are cited. For the model built, our team has been fully justified based on the data from the questionnaire results and presented in the form of tables or textual analysis.
Point 5: The discussion section also should be more expanded. For example, what are the factors which influence environmental education and how government can contribute to it.
Response 5: In the conclusion section, we have regrouped and changed the title of the conclusion section to Conclusions and Discussions. In the content section, we also made a lot of content filling to fully explain the role of environmental education in environmental governance performance. At the same time, we also give corresponding suggestions to the government, while calling on people to take action to protect the environment and respect nature. Echoing the theme, we believe that this paper has some educational significance.

Reviewer 2 Report
The submitted work represents a good addiction to scientific literature, but an additional background information is needed to help no experts in the field and to increase the overall understanding of the importance of the paper. Details are below.
Some typos occur. Check for “-“ inside words, and others.
I am not sure the paper can be considered as a “research paper”, while, probably, the “review” tag is more suitable.
“Conclusions and discussion” sound better than recommendations.
The introduction needs to introduce a bit more each single environmental pollutant. This part I suggest to add is really important for a reader to have better idea on what the pollutants are. A few lines and references supporting each pollutant should be added in order to help readers know also pollutants that they eventually are not expert on. For my personal experience, I can suggest a period about noise pollution, which is a relevant and major concern. The authors can use it also as a suggestion to fetch online database for the other pollutants.
“Noise pollution is scientifically recognized as environmental pollutant associated to sleep disorders (Muzet A. Environmental noise, sleep and health. Sleep Med Rev 2007; 11: 135–42), learning impairment (Zacarías, F. F., Molina, R. H., Ancela, J. L. C., López, S. L., & Ojembarrena, A. A. (2013). Noise exposure in preterm infants treated with respiratory support using neonatal helmets. Acta Acustica united with Acustica, 99(4), 590-597; Erickson, Lucy C., and Rochelle S. Newman. "Influences of background noise on infants and children." Current Directions in Psychological Science 26.5 (2017): 451-457.), hypertension ischemic heart disease (Babisch, W., Beule, B., Schust, M., Kersten, N., Ising, H., ‘Traffic noise and risk of myocardial infarction’, Epidemiology, 16, 2005, pp. 33–40. ), diastolic blood pressure (Petri, D., et al. (2021). Effects of Exposure to Road, Railway, Airport and Recreational Noise on Blood Pressure and Hypertension. Int. J. Environ. Res. Public Health 2021, 18(17), 9145), reduction of working performance (Vukić, L., et al. (2021). Seafarers’ Perception and Attitudes towards Noise Emission on Board Ships. International Journal of Environmental Research and Public Health, 18(12), 6671. Rossi, L., Prato, A., Lesina, L., & Schiavi, A. (2018). Effects of low-frequency noise on human cognitive performances in laboratory. Building Acoustics, 25(1), 17-33.), annoyance (Miedema HME, Oudshoorn CGM. Annoyance from transportation noise: relationships with exposure metrics DNL and DENL and their confidence intervals. Environ Health Perspect 2001; 109: 409–16.). Major transportations are the sources resulting to be the most impactful on human life style: road traffic (Cueto, J. L., Petrovici, A. M., Hernández, R., & Fernández, F. (2017). Analysis of the Impact of Bus Signal Priority on Urban Noise. Acta Acustica united with Acustica, 103(4), 561-573. Fredianelli, Luca, et al. "Traffic flow detection using camera images and machine learning methods in ITS for noise map and action plan optimization." Sensors 22.5 (2022): 1929. Ruiz-Padillo, Alejandro, et al. "Selection of suitable alternatives to reduce the environmental impact of road traffic noise using a fuzzy multi-criteria decision model." Environmental Impact Assessment Review 61 (2016): 8-1), railway traffic (Bunn, Fernando, and Paulo Henrique Trombetta Zannin. "Assessment of railway noise in an urban setting." Applied Acoustics 104 (2016): 16-23), airports (Iglesias-Merchan, Carlos, Luis Diaz-Balteiro, and Mario Soliño. "Transportation planning and quiet natural areas preservation: Aircraft overflights noise assessment in a National Park." Transportation Research Part D: Transport and Environment 41 (2015): 1-12;), port activities (Fredianelli, Luca, et al. "Source characterization guidelines for noise mapping of port areas." Heliyon (2022): e09021; Nastasi, Marco, et al. "Parameters affecting noise emitted by ships moving in port areas." Sustainability 12.20 (2020): 8742.). are the most diffused ones”. This are also major sources for air pollution … Please also consider there are cross studies between air and noise pollution if possible.
Author Response
Response to Reviewer 2 Comments
The submitted work represents a good addiction to scientific literature, but an additional background information is needed to help no experts in the field and to increase the overall understanding of the importance of the paper. Details are below.
Point 1: Some typos occur. Check for “-“ inside words, and others.
Response 1: For this issue, we are uploading the article according to the template specified by the journal, that's why the problem occurs, it's the end of line words to be hyphenated because we applied the two end alignment format. In addition to such a problem at the end of the sentence, we checked the middle part of the sentence in the article and corrected it in time.
Point 2: I am not sure the paper can be considered as a “research paper”, while, probably, the “review” tag is more suitable.
Response 2: This paper empirically analyzes the impact of environmental education on environmental governance performance, using public participation as a mediator. A questionnaire was distributed, a model was developed, and a reliability analysis was conducted based on the data using software, followed by a full justification. Thus in summary, I believe that our paper is more appropriately considered as ”research paper”.
Point 3: “Conclusions and discussion” sound better than recommendations.
Response 3: In the conclusion section, we have regrouped and changed the title of the conclusion section to Conclusions and Discussions. In the content section, we also made a lot of content filling to fully explain the role of environmental education in environmental governance performance. At the same time, we also give corresponding suggestions to the government, while calling on people to take action to protect the environment and respect nature. Echoing the theme, we believe that this paper has some educational significance.
Point 4: The introduction needs to introduce a bit more each single environmental pollutant. This part I suggest to add is really important for a reader to have better idea on what the pollutants are. A few lines and references supporting each pollutant should be added in order to help readers know also pollutants that they eventually are not expert on. For my personal experience, I can suggest a period about noise pollution, which is a relevant and major concern. The authors can use it also as a suggestion to fetch online database for the other pollutants.
Response 4: We have taken your suggestions fully into account, and in order to give the reader a better understanding of what a contaminant is, we have added an introduction to the various contaminants in the introduction, and added corresponding references. In particular, we have adopted your suggestion to add the aspect of noise pollution and cited the references you mentioned.
For all modifications, we have already marked tu'chu'xu in red font in the article.
Point 5: Please also consider there are cross studies between air and noise pollution if possible.
Response 5: Regarding your question about increasing the research between air and noise pollution, due to the limitations of our current research, we are unable to conduct a more in-depth study in this paper. However, in future studies, we will take your suggestion into full consideration and continue to explore it in depth in our next studies.

Round 2
Reviewer 1 Report
Accept
Reviewer 2 Report
The paper is much better and ready for being published.